# Development of a Broadband (100–240 MHz) Surface Acoustic Wave Emitter Devoted to the Non-Destructive Characterization of Sub-Micrometric Thin Films

**DOI:** 10.3390/s22197464

**Published:** 2022-10-01

**Authors:** Marc Duquennoy, Nikolay Smagin, Tahar Kadi, Mohammadi Ouaftouh, Frédéric Jenot

**Affiliations:** IEMN (UMR CNRS 8520), University Polytechnique Hauts-de-France, CNRS, University Lille, F-59313 Valenciennes, France

**Keywords:** chirped interdigital transducers, surface acoustic waves, SAW dispersion, very high frequency (VHF), thin films

## Abstract

In the ultrasonic non-destructive evaluation of thin films, it is essential to have ultrasonic transducers that are able to generate surface acoustic waves (SAW) of suitably high frequencies in a wide frequency range of between ten and several hundred megahertz. If the characterization is carried out with the transducer in contact with the sample, it is also necessary that the transducers provide a high level of mechanical displacement (>100 s pm). This level allows the wave to cross the transducer–sample interface and propagate over the distance of a few millimeters on the sample and be properly detected. In this paper, an emitter transducer formed of interdigitated chirp electrodes deposited on 128° Y-cut LiNbO_3_ is proposed. It is shown that this solution efficiently enables the generation of SAW (displacement level up to 1 nm) in a frequency range of between 100 and 240 MHz. The electrical characterization and a displacement field analysis of SAW by laser Doppler vibrometry are presented. The transducer’s significant unidirectionality is demonstrated. Finally, the characterization of two titanium thin films deposited on silicon is presented as an example. A meaningful SAW velocity dispersion (~10 m/s) is obtained, which allows for the precise estimation (5% of relative error) of the submicrometer thickness of the layers (20 and 50 nm).

## 1. Introduction

Many fields of engineering use thin film technology, and the semiconductor industry, in particular, accounts for a significant portion of their use. The manufacturing of microelectronic devices is entirely dependent on the ability to successively deposit the thin films of materials according to closely controlled specifications. The continued miniaturization of electronics thus depends on the ability to produce thin films at increasingly small-scale factors but with the same level of control over the structure and properties of the film. Consequently, the accurate measurement of thin film parameters at each stage of fabrication is very important for controlling structures using thin films. This is also true for thin films used outside the semiconductor industry, where the accurate measurement of the properties of thin films is generally necessary in order to validate their functional performance. 

Although thin film deposition is not usually performed for structural reasons, the mechanical characterization of the film is often desired. The layer becomes the point of physical contact with the external environment. Whether or not the mechanical properties serve the desired functional quality is secondary to the fact that the mechanical behavior will govern the reliability of the thin film when it is put into service. For example, in the semiconductor industry, the mechanical properties of layers may not influence their electrical or thermal performance. However, a certain degree of mechanical reliability is usually required in order to withstand the polishing and packaging steps during manufacturing. Thus, the accurate measurement of the mechanical properties of the layers deposited at each step is necessary in order to develop the post-processing parameters for the subsequent steps and the functionalization of the final product. The main challenge regarding the mechanical characterization of thin films is the scale on which the test has to be performed. On a micrometer or nanometer scale, traditional techniques such as tensile or beam bending tests are simply not feasible. Scaling up these tests to the thickness of the layer may seem to provide a viable solution by enabling the testing of the layer on a substrate. However, this inevitably results in the coupling of the mechanical properties of the substrate and the thin layer. To accurately measure the mechanical properties of the layer, the test must somehow be localized on the layer, or a means of decoupling the influence of the substrate on the results must be found. 

Another challenge, especially for nanosized layers, is that the properties of the layer can vary significantly from those of the same bulk material. Just as the hardness of bulk materials can vary with the grain size, thin films can exhibit different behaviors at decreasing thicknesses. Given that semiconductor manufacturing is now carried out with layer thicknesses of the order of tens of nanometers, a broad understanding of the material behavior on this scale is necessary in order to accurately measure the properties of thin films. The above challenges highlight the difficulty of characterizing thin films based on geometric constraints (thickness, nanostructure). However, another troublesome constraint may emerge from the morphology of the thin film. 

There are several techniques used for characterizing thin films, but nanoindentation is by far the most widely used, as the sample requires no special preparation. It is fast, and various types of thin films can be tested. The technique is essentially a macro-indentation test that involves the testing of the sample using a sharp diamond tip. However, unlike large-scale tests, the elastic properties can be measured from the initial elastic response of the unloading segment, and the accuracy of the measurement is dependent on the depth of penetration of the indenter into the layer. Furthermore, the stiffness of the layer, with respect to the stiffness of the substrate, has to be taken into account [1,2,3,4].

An alternative means of non-destructive characterization is found in the use of ultrasonic methods. In this work, the use of Rayleigh-type surface acoustic waves (SAW) were favored [5]. They consist of longitudinal and shear displacements coupled together, which propagate at the same velocity. Both components are in phase quadrature, so that the polarization plane is elliptical [6]. In particular, the rotation of the displacement vector is counterclockwise (retrograde) at the material surface and clockwise (progressive) below the surface. The amplitude of the vibrations caused by the SAW is higher at the material’s surface and decreases exponentially in the material. At the depth of a few wavelengths, it reduces to 1/*e* compared to the value on the surface (ln€ = 1). This distance from the surface of the material is defined as the penetration depth of the SAW. The higher the frequency is, the more the SAW energy is concentrated in a thin layer, starting at the sample’s surface. This fact makes SAWs very sensitive to the mechanical characteristics of the surface and the near-surface region. Therefore, high-frequency SAW is a powerful non-destructive tool for the characterization of thin films deposited on a substrate whose thickness may be much thinner than the penetration depth of the wave. While for a completely homogeneous sample the SAW velocity is constant, SAW propagation of a thin film deposited on a substrate is dispersive, because the SAW velocity is a function of both the SAW frequency and the thickness of the thin film, and also the elastic properties of the film. 

There are several methods used to generate SAW, but when characterizing thin films, some physical constraints arise (the dimensions of the samples, nature of the material of the thin film (metallic, polymer), anisotropy, etc.) and reduce the choice of methods. Several techniques have been used to generate Rayleigh-type surface waves, including laser-ultrasound [7,8,9], wedges [10], immerged transducers [11], line-focus acoustic transducers [12] and interdigital transducers (IDTs) [13,14,15]. The latter enable broadband SAW generation on a piezoelectric substrate in an easily controllable manner using a chirped electrode finger configuration [16]. Since IDTs are offset transducers used for NDT applications, it is necessary to provide sustainable acoustical contact with the structure under testing. This issue is sometimes circumvented by integrating IDT electrodes and the piezoelectric substrate directly into the structure [17,18] but with unavoidable limitations regarding the configurations of the samples to be characterized.

Initially appearing in telecommunications, where they are successfully used in the frequency range of up to a few GHz, IDTs have subsequently found numerous applications in microfluidics, sensors, and non-destructive testing. The latter have in common the fact that the acoustic waves interact mechanically with the environment. Considering that, in this case, SAWs often have to undergo significant attenuation, cross different solid–solid and solid–liquid interfaces, create energy input and undergo other influences, the operating range of IDTs is limited to a few hundred MHz [19].

In the presented paper, we use a chirped (or dispersive) IDT as a SAW emitter placed in acoustic contact with a layer-on-substrate test sample. Contact measurements have the advantage of being non-destructive and compatible with all types of coatings, such as fragile or transparent films. The transducer operating frequency band is 100–240 MHz. It is shown by laser Doppler vibrometry (LDV) that it can provide a SAW displacement level of up to 1 nm, which enables the probing SAW to cross the transducer–sample interface, propagate through several millimeters and be optically detected. The chosen frequency range is high enough to detect a SAW dispersion of the order of 10 s m/s and enable submicron (10 s of nm) film characterization. 

## 2. Broadband (100–240 MHz) Surface Acoustic Waves Emitter

The broadband surface acoustic waves emitter used in this study is based on an interdigital transducer formed of interdigitated chirp electrodes deposited on 128° Y-cut LiNbO_3_. Interdigital transducers consist of comb-shaped metal electrodes (often made of gold or aluminum) composed of interlocking fingers with a finger overlap distance of *W* [20,21,22]. The IDT electrode structure is presented at the top of Figure 1a for an IDT with a constant electrode period. The electrodes are deposited on piezoelectric substrate so that, when an electrical voltage is applied between the two adjacent electrodes, an accumulation of charges is created, whose signs alternate from one finger to the other. This implies the creation of an electric field between each pair of fingers. The combination of the piezoelectric properties of the substrate and this field leads to expansions and compressions in the material, creating displacements [22]. When the applied electric voltage *V*_E_ is sinusoidal, vibrations are constructively created only if the periodicity *p* is equal to half a wavelength of the Rayleigh wave λ, thus producing surface acoustic waves. In the case of the periodic structure, they are emitted on both sides of the transducer (SAW_±_ in the IDT side, viewed at the bottom of Figure 1a). Thus, the frequency *f*_0_, which corresponds to this cumulative effect, is called the synchronous frequency or resonant frequency, defined as *f*_0_ = *V*/2*p*. Here, *V* is equal to the propagation velocity of the SAW in the substrate and *p* is equal to the periodicity of the interdigital electrodes, and the acoustic wavelength λ is equal to double the value of the period λ = 2*p*. Due to the reciprocity of the piezoelectric field, it is possible to acquire the propagated wave in an electrical form using another interdigital transducer of the same type. The signal received in this way is delayed and attenuated in relation to the transmitted signal, and this type of configuration makes it possible, for example, to create filters or delay lines.

Figure 1b shows a schematic diagram of the chirped (or dispersive) SAW emitter used in the present study. It is fabricated using a standard photolithography process on LiNbO3-Y+128° followed by lift-off. It is composed of 414 gold electrodes, whose periodicity is variable to ensure the linear frequency modulation of the emitted signal. The maximum electrode width *a* (*a* = *p*/2) is 11 µs, the minimum is 3.8 µm and the thickness *h* = 450 nm. Thus, the electrode-thickness–period ratio varies between approximately 2 and 6%. The metallization ration is 50%. The aperture of the IDT is *W* = 2.5 mm and its length is *L* = 4 mm. The ratio between the aperture *W* and the minimum wavelength λ_min_ is equal to approximately 56, which prevents the diffraction of the emitted acoustic field at propagational distances large enough for the layer characterization. The final IDT, mounted on a PCB with its SMA connector plug, is shown in Figure 1c. The electrical connection with the IDT is assured by soldered gold microwires. 

The electrical response of the transducer was measured using a manual RF probe station with a vector network analyzer (Rohde&Schwartz ZND). Figure 2 shows the results of the measurement. The curve obtained is a coherent superposition of the responses of different electrode pairs corresponding to different frequencies [23]. With the increasing frequency, the resistance increases and reaches a value of 8 Ohms at 215 MHz. The increase in the resistance is due to the fact that the number of electrodes increases quadratically as a function of the instantaneous frequency in order to ensure the linear frequency modulation [24].

The displacement magnitude of the surface acoustic waves was measured optically using the Polytec UHF-120 laser vibrometer. The measurement point was located on the centerline of the IDT at a distance of 2 mm from the transducer aperture. In Figure 1b, the measurement point is shown with a blue circle. The IDT electrical excitation was chosen according to the dual temporal-spatial chirp excitation method [21]. It represented a linear chirp descending from 240 to 100 MHz, having a duration of 1 µs and an amplitude of 20 V_0-p_. The mentioned IDT length (1 mm) determines that the duration of the emitted chirped pulse is equal to around 2 µs, given that the Rayleigh velocity is 3878 m/s for LiNbO3-Y+128°. Thus, the time–frequency product (TFP, product of the pulse duration and signal bandwidth) is equal to approximately 280. Designing chirped transducers with a TFP in the range of 100–300 is practical regarding the need to ensure the minimization of the IDT dimensions and the level of Fresnel ripples in the emitted spectrum while maintaining a high emitting efficiency [24,25].

The waveform of the IDT emission signal is shown in Figure 3a. The geometrical configuration of the transducer and its mode of excitation causes the high-frequency vibrations to appear at the beginning of the wave packet and drop to lower frequencies according to the relations of linear frequency modulation. The shape of the wave packet envelope presents numerous local maxima and minima. Its instability is more pronounced at high frequencies. The level of displacement of the surface acoustic waves is up to one nanometer.

The spectrum of the wave packet is shown in Figure 3b. It corresponds to a typical spectrum of a linear frequency modulated signal. The oscillations of the spectrum observed in the 100–240 MHz generation frequency band correspond to ripples in the envelope of the temporal signal (Figure 3a). However, it can be concluded that the acoustic energy is present in the whole frequency range from 100 to 240 MHz, which is sufficient for obtaining the SAW dispersion curves of the layered structures under investigation.

To identify the reasons behind the ripples observed in the time signal envelope, an optical B-Scan in the longitudinal direction of the transducer was performed. The scan line is located in the middle of the transducer and slightly exceeds the limits of the electrode overlap area. It is represented by green squares in Figure 1b. The scan result is shown in Figure 3c. This space–time diagram displays the bidirectional emissions of the elementary interdigital transducer electrode pairs. In correspondence with the excitation chirp signal, the emission starts at the maximum frequency of 240 MHz at a time instant of 3.5 µs. The other electrode pairs are subsequently activated, progressing linearly from high to low frequencies. The emission disappears at approximately 4.5 µs. 

The negative direction of the *x*-axis shown in Figure 3c (see also Figure 1b) corresponds to the emission towards the backside of the IDT, which is not used for characterizations. It can be observed that the high-frequency waves do not penetrate through the electrode grid so as to be emitted in the reverse direction of the transducer. This fact can be explained by the high reflectivity of the 450 nm-thick gold electrodes used for high-frequency waves [26]. According to our numerical simulations, the electrodes’ reflectivity per period varies from 2.5% to −9.5% as the *h*/*p* ratio increases from 2% to 6%. 

The superposition of the waves emitted by the pairs of elementary electrodes forms the final wave packet emitted by the transducer in the positive direction of the *x*-axis. The duration of the acoustic wave packet (2 µs) is twice that of the electrical excitation signal. Significant interference in the overlap area of the electrodes is observed. This is responsible for the distortions of the temporal signal envelope visible in Figure 3a. The A-scan, shown in Figure 3a, is taken as the point where the *X* coordinate = 4.5 mm, according to Figure 3c. 

The spatial spectrogram presented in Figure 4 offers greater insight into the directivity of the dispersive IDT emission. It represents the results of the FFT for the B-scan data in Figure 3c. For example, in [25,27], directional emission was simulated using P-matrix formalism for a series of chirped transducers deposed on LiNbO_3_ and possessing 200 nm-thick aluminum electrodes. The difference in the acoustic power, emanating in opposite directions, was demonstrated. However, all the frequency components were present in the forward- and backward-emitted SAW. At the same time, our experimental results show that the reflectivity of a gold electrode grating causes all the high-frequency components to be completely redirected in the direction of forward emission (Figure 4). Only a narrow band signal in the vicinity of 100 MHz is able to propagate in the backward direction. The spectrum energy ratio between the forward and backward emissions is about 7.8 dB.

The nonuniformities caused by interference and reflection seen in Figure 3 and Figure 4 do not perturb the characterization process. As shown in Section 3 below, the slant-stack transform, used to reconstruct the dispersion curves, operates in association with each frequency component individually and relative to its initial state. Thus, for the characterization to be successful, we need only a sufficient amount of energy for a given frequency range, but the uniformity of its distribution does not matter. As the slant-stack provides a relative comparison of the wave packets, only the perturbations due to propagation will be detected. The results of the dispersion curve calculations shown in Section 3 (Figure 7) confirm this statement. 

Additionally, the complete spatial–temporal distribution of the wave packet was optically measured by performing a B-scan in the direction perpendicular to the acoustic propagation (Figure 5). The scan line was located in the vicinity of the IDT aperture (shown by the red diamonds in Figure 1b). The colormap corresponds to the displacement in nanometres. The laser Doppler vibrometry shows a stable emission over the whole aperture width and for all the emitted frequencies.

In conclusion, we demonstrated that the Rayleigh SAW emitter can provide a sufficient level of acoustic vibrations (up to 1 nm) in the 100–240 MHz range. The emitted wave packet does not exhibit any wavefront distortions (Figure 5). However, significant ripples in the signal envelope above 200 MHz and a pronounced unidirectionality (7.8 dB) can be detected using laser Doppler vibrometry. These effects are due to the strong reflectivity of the gold electrode grating. Using thinner electrodes made of aluminum would decrease the level of reflectivity, but at the same time, the level of directivity would also be reduced. Overall, the total amount of acoustic energy that the emitter can direct in the case of the test sample may be lower compared to an IDT with gold electrodes. As the energy distribution uniformity is not considered an important factor, the use of less reflective electrodes could represent a disadvantage. This statement must be proved by further comparative study using a SAW emitter made of aluminum electrodes.

## 3. Application to the Characterization of Thin Films

To show the benefits of such a transducer, the dispersion of the SAW propagation velocities was measured for several layer-on-substrate structures. The measurements were carried out at the IEMN’s WAVESURF platform (Figure 6). The device is primarily composed of an interferometer that enables the detection of out-of-plane SAW displacements on the surface of the controlled samples [28].

The characterization using Laser Doppler vibrometry was achieved using a commercial scanning heterodyne Polytec UHF-120 vibrometer. The apparatus enables measurements in the DC within an ultra-high frequency (1.2 GHz) range. The scanning resolution is defined according to the motorized stages and is equal to 300 nm. According to the device data sheet, its SAW amplitude sensitivity threshold, which depends on the signal-to-noise ratio (SNR), is defined as 30 fm √(Hz). This gives a low value of 268 pm for a single measurement at 100 MHz and dictates the necessity of averaging. Thus, the sensitivity can be improved proportionally to the square root of the average number of sweeps. In all the experiments reported below, 128 signals were averaged, resulting in a noise level value of 23 pm. 

To demonstrate the very high sensitivity of SAWs to the presence of extremely thin films or layers, two layer-on-substrate structures with 20 and 50 nm-thick titanium thin films on a silicon substrate were studied. As these frequencies are in the 100 MHz range, the attenuation is not too strong (unlike the attenuation of waves propagating at frequencies of around 1 GHz). It is then possible to propagate waves over several tens of millimeters, record the associated displacements over a sufficient length and obtain accurate results for the experimental dispersion curves. For the measurements, 927 detection points were recorded, with a step size of 35.6 µm between each point. The experimental dispersion curves were obtained using the slant-stack transform [29].

There are several methods used to obtain the phase velocity dispersion, and when considering the exploitation of the dispersion phenomena of the first Rayleigh mode, it is possible, for example, to apply a wavelet transform to signals corresponding to different propagation distances [30,31,32]. Phase velocities can also be obtained using a two-dimensional fast Fourier transform applied to time signals recorded at uniformly spaced intervals [33,34]. In this work, we decided to use a fast and efficient alternative method to estimate the velocities with a high accuracy. Being well known in geophysics, the slant-stack transform is used to determine the velocity of waves propagating in a dispersive medium [35,36,37]. In terms of precision, the results obtained using this method are as good as those obtained using the other methods (2DFFT, wavelet) [29], while also providing the possibility of observing the dispersion curves directly. The slant-stack method allows for the precise determination of the phase velocity dispersion curves from the displacements measured at different spatially distributed points. This is known as the *p*-ω transformation or oblique summation. Signal processing using this transform enables the phase velocity to be determined as a function of frequency. The method is useful, in particular, for the case of Rayleigh waves [38,39]. The procedure for measuring the velocity *V*_R_ of Rayleigh waves as a function of frequency consists in calculating the maximum of the function *A*(*V*) at the output of a correlator that measures and compensates for the delays between the signals detected by the vibrometer at positions 1, 2 and *N*. In other words, a velocity band is defined such that *V*_min_ < *V*_R_ < *V*_max_, and then *A*(*V*) is determined. *A*(*V*) is a function of the velocity *V*; therefore, the velocity at which the function is at its maximum corresponds to the measured wave velocity for the frequency *f*_0_ under consideration. The function *A*(*V*) is defined by [36]:$$A{\left(V\right)}_{\langle V,f_{0}\rangle }=S\left(f_{0}\right)N\left|\frac{\sin\left(N\cdot \Delta x\pi f_{0}\left(\frac{1}{V}-\frac{1}{V_{R}}\right)\right)}{\left[N\cdot \sin\left(\Delta x\pi f_{0}\left(\frac{1}{V}-\frac{1}{V_{R}}\right)\right)\right]}\right|,$$
where Δ*x* is the distance between the measurement points, *f*_0_ is the frequency, *V*_R_ is the measured wave velocity, *N* is the number of measurement points, *V* is the wave velocity (considered as a variable) and *S* is the constant resulting from the cross-correlation of all the signals received.

The experimental dispersion curves were determined using the slant-stack transform applied to two wafers covered with approximately 50 and 20 nm-thick gold layers deposited on a silicon substrate. In Figure 7, the experimental dispersion curves (solid lines) with chromium thicknesses of 20 and 50 nm for the wave propagation in the [$1\bar{1}$0] direction are presented, indicating the possibility of characterizing the titanium layer thicknesses. By observing these experimental dispersion curves and considering their quality, we can see that it is possible to perform the inversion. The velocity variations observed were of the order of 10 to 30 m/s depending on the specific case.

The estimation of the thickness of the layers *E*_p_ using the experimental measurements of the velocity dispersion consisted in solving the inverse problem. The dispersion curves depend, in particular, on the thickness of the layers, as well as the Young’s modulus *E* and Poisson’s ratio ν of the layers and the substrate. For each coated wafer, an inversion method was used to identify the best of the layer thickness fit (the elastic coefficients of the layer and substrate are known). This enables dispersion curves of the theoretical and experimental velocities that are as close as possible to be obtained. First, the theoretical calculation was carried out using rounded values for the thicknesses. These theoretical phase velocity dispersion curves were calculated within the 100–240 MHz band. Then, a least squares minimization routine was performed to adjust the set of parameters, so that the theoretical velocities were as close as possible to the measured velocities. Thus, the inversion algorithm is based on the dichotomy principle and the minimization of the Pearson coefficient *R*^2^. The objective function *R*^2^ must be as close to 1 as possible.

The inversion results yielded thicknesses of 45 nm and 21 nm for the two layers, with *R*^2^ = 0.9964 and 0.9993, respectively. This is quite satisfactory, considering that there is always a difference between the set points imposed on the evaporation racks and the actual thicknesses deposited. These values were compared with those obtained using a profilometer and the differences were of the same order of magnitude (±2 nm). As the measurement errors are often estimated as being within a few percent, these results corroborate the relevance of the values obtained using ultrasound. It is also important to specify that thickness measurements using a profilometer can only be carried out for one step intentionally manufactured on the sample and their accuracy decreases if the measurement is made over a long distance, whereas ultrasonic measurements can be made over the entire surface of the wafer, including in the center of the sample

## 4. Conclusions

In this paper, a broadband surface acoustic waves emitter based on an interdigital transducer formed of interdigitated chirp electrodes deposited on 128° Y-cut LiNbO_3_ was presented. Implementing a piezoelectric material such as LiNbO_3_ with an important electromechanical coupling coefficient [40] allowed us to obtain a high SAW displacement level (up to 1 nm) in the 100–240 MHz frequency range. Additionally, using relatively thick (450 nm) and highly reflective gold electrodes assures the high unidirectionality of the transducer, which efficiently enables the redirecting of the acoustic energy into the non-destructively tested sample (crossing of the transducer–sample interface by the wave) and obtain its propagation on the sample over a distance of a few millimeters. The presented SAW emitter, which is based on dispersive IDT, permitted us to estimate the thickness of the 20 and 50 nm Ti layers with a 5% relative error. Future research will involve further increasing the transducer frequency in order to obtain a better characterization precision. However, a comprehensive analysis of the limiting phenomena, such as the decreased SAW displacement level and increased attenuation, will be necessary.

## Figures and Tables

**Figure 1 sensors-22-07464-f001:**
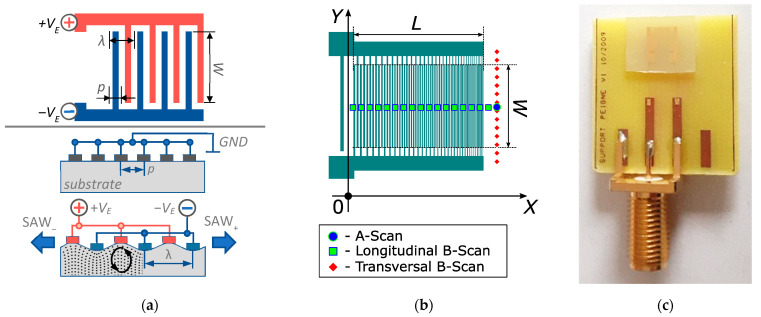
(**a**) SAW emission principle with a constant-period IDT. (**b**) Schematic image of the chirped IDT SAW emitter and locations of optical measurement points. (**c**) Photo of the chirped IDT used for thin film characterization.

**Figure 2 sensors-22-07464-f002:**
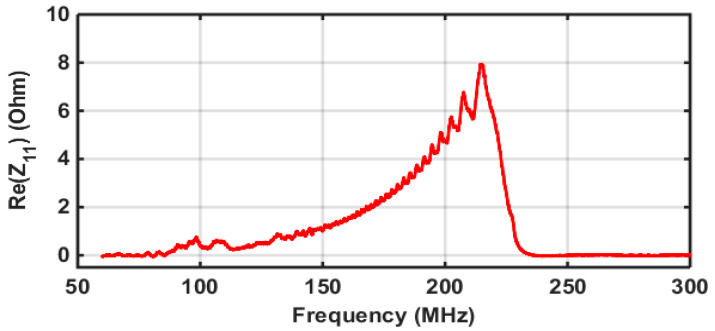
Electrical resistance of the chirped IDT transducer.

**Figure 3 sensors-22-07464-f003:**
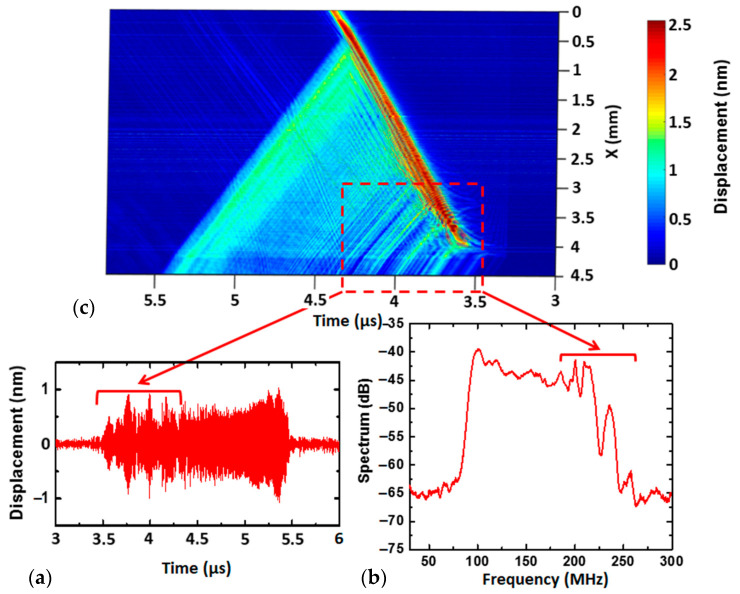
Optical measurement of SAW emission by the chirped IDT: (**a**) A-scan, corresponding to the displacement of the surface wave in the time domain; (**b**) spectrum of the same signal; (**c**) longitudinal B-Scan, revealing bidirectional emission, multiple reflections and the formation of the temporal envelope of the emitted wave packet.

**Figure 4 sensors-22-07464-f004:**
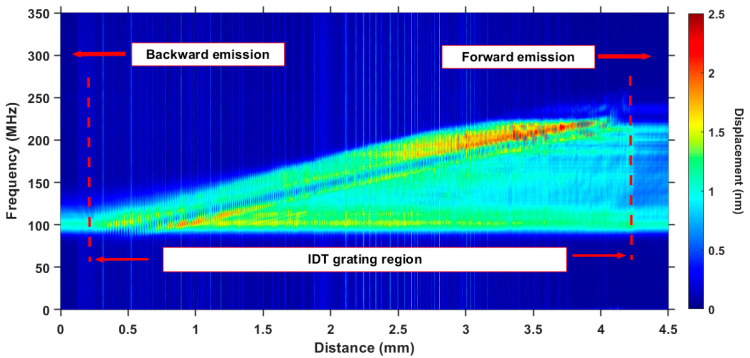
Optical measurement of the SAW emission by the chirped IDT: FFT of the longitudinal B-scan.

**Figure 5 sensors-22-07464-f005:**
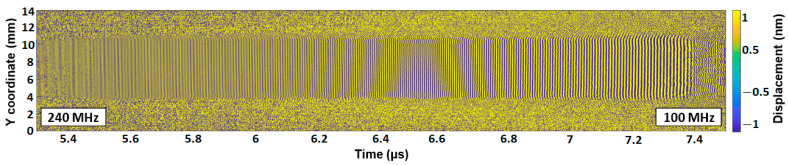
Optical measurement of the SAW emission by the chirped IDT: transversal B-scan showing the generated SAW packet.

**Figure 6 sensors-22-07464-f006:**
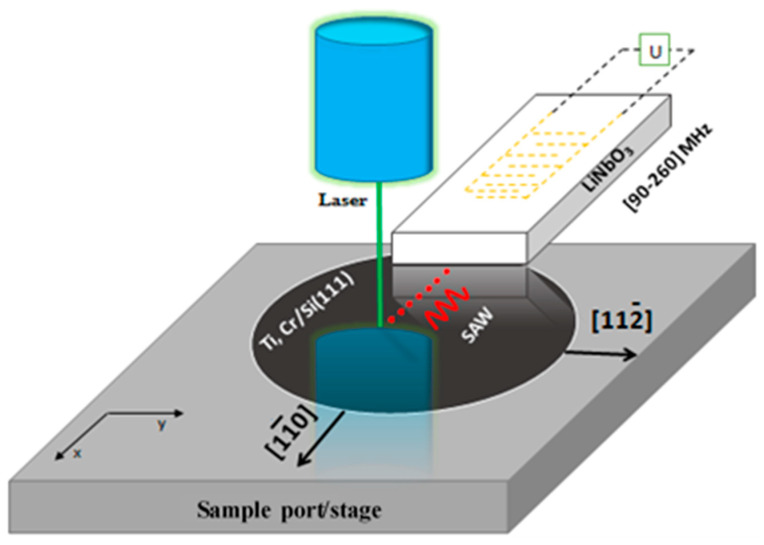
Schematic diagram of the measuring principle.

**Figure 7 sensors-22-07464-f007:**
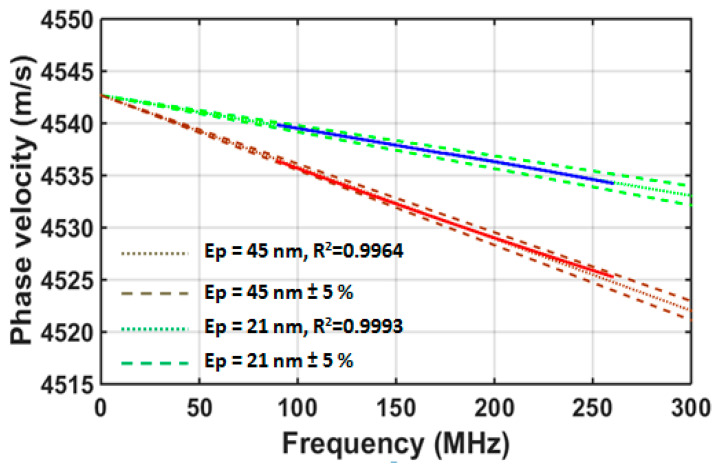
Theoretical (dotted lines) and experimental (solid lines) dispersion curves for 20 and 50 nm of titanium on silicon in the [1$\bar{1}$0] direction, and the thickness *E*_p_ inversion results. Dashed lines are theoretical curves corresponding to a ±5% thickness variation relative to the initial estimated value.

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
