# Peer review of "Development of a Broadband (100–240 MHz) Surface Acoustic Wave Emitter Devoted to the Non-Destructive Characterization of Sub-Micrometric Thin Films"

_sensors, 2022, doi:10.3390/s22197464_

Round 1
Reviewer 1 Report
There are many corrections that must be done by the authors. I have included all comments in the marked/commented manuscript as attached.
I also found four (04) self-citations which require further attention of the editors.

Author Response
Please see the attachment:

Reviewer 2 Report
Please reconsider the title of the paper. Interdigital transducers (IDTs) are well known as the effective structures for excitation and detection of surface acoustic waves (SAWs) of different types. The mentioned frequencies 100-240 MHz have been high 20 years ago. It would by much better to focus on description of deposited surface layer tests results treating the IDT and SAW as tools only. SAW related phenomena are well known but nondestructive surface testings are still interesting. The part of the paper devoted to the measurements results are very valuable, in my opinion (especially Fig. 4).
Please take into consideration that Rayleigh waves are one of several different types of surface acoustic waves. The sentence "Surface acoustic waves (SAW), also called Rayleigh waves,[...]" perpetuates only colloquial nomenclature.
A broadband IDT you show in Fig.1 is called in the literature "dispersive transducer". Usually the term "broadband IDT" is used for very short IDTs.
Please use italic fonts and correct subscripts close to the formula A(V).
The conclusions (and whole article) should be focused on measurement results of surface thin layer. The IDTs and SAWs properties are well known for several decades.
Author Response
Please see the attachment:

Reviewer 3 Report
Generation of SAW with IDTs is a common practice in implementing SAW filters (ghost image rejection, convolvers, matched filters..) up to 3 GHz. Please see the website below. This manuscript, however has some interesting contributions to SAW surface displacement imaging. I would change the title to "Real-time imaging of SAW Generated Surface displacement..." and would remove "effective" and "interdigital transducers" since those are known for more than 70 years.
https://www.golledge.com/products/saw-filters-from-golledge-electronics/c-26/c-81?gclid=Cj0KCQjw08aYBhDlARIsAA_gb0cGkrvsrMIbNF-uVxFBU81RAySAezRQfVhYHJCF7R6LisKWAuUvqpMaAqtgEALw_wcB
Author Response
Please see the attachment:

Round 2
Reviewer 2 Report
Thank you very mauch for the changes. Now the topic and content of the paper fit together.
Author Response
Thank you very much for reviewing our paper.